# Fasting-induced activity changes in MC3R neurons of the paraventricular nucleus of the thalamus

Robert A Chesters[1,*], Jiajie Zhu[1,3,*], Bethany M Coull[1,3], David Baidoe-Ansah[1,3], Lea Baumer[1], Lydia Palm[1], Niklas Klinghammer[1], Seve Chen[1], Anneke Hahm[1], Selma Yagoub[1], Lídia Cantacorps[1,2], Daniel Bernardi[1], Katrin Ritter[1], Rachel N Lippert[1,2,3]

The brain controls energy homeostasis by regulating food intake through signaling within the melanocortin system. Whilst we understand the role of the hypothalamus within this system, how extra-hypothalamic brain regions are involved in controlling energy balance remains unclear. Here we show that the melanocortin 3 receptor (MC3R) is expressed in the paraventricular nucleus of the thalamus (PVT). We tested whether fasting would change the activity of MC3R neurons in this region by assessing the levels of c-Fos and pCREB as neuronal activity markers. We determined that overnight fasting causes a significant reduction in pCREB levels within PVT-MC3R neurons. We then questioned whether perturbation of MC3R signaling, during fasting, would result in altered refeeding. Using chemogenetic approaches, we show that modulation of MC3R activity, during the fasting period, does not impact body weight regain or total food intake in the refeeding period. However, we did observe significant differences in the pattern of feeding-related behavior. These findings suggest that the PVT is a region where MC3R neurons respond to energy deprivation and modulate refeeding behavior.

## Introduction

The progression of obesity, diabetes, and other metabolic disorders in our society is occurring at an alarming speed, and there is a great need to understand the underlying physiological changes that are occurring. Current preclinical animal models used to study obesity rely on either genetic models or, more frequently, long-term exposure to unhealthy diets across weeks/months (Lutz & Woods, 2012; Suleiman et al, 2020; de Moura E Dias et al, 2021). These studies highlight that it is the gradual accumulation of small behavioral, physiological, and molecular changes that shift energy balance in a positive direction, leading to an increased likelihood of obesity (Schwartz et al, 2017). The control of energy balance is predominately regulated through changes in the brain (Cone, 1999, 2005; Ellacott & Cone, 2004; Dietrich & Horvath, 2013).

The primary regulating system of energy balance in the brain is the central melanocortin system (Cone, 2005). Agouti-related peptide (AgRP) and proopiomelanocortin (POMC) neurons reside in the arcuate nucleus (ARC) of the hypothalamus and send projections to brain regions expressing melanocortin receptors to mediate aspects of energy balance. Almost all studies related to fasting or feeding responses have focused on this system and its activity directly within the hypothalamus.

A number of recent studies centered on this system have focused on the rapid neuronal changes that occur in response to immediate feeding-related actions such as the sight, smell and taste of food, immediately after a fasting period (Chen et al, 2015; Chen & Knight, 2016; Brandt et al, 2018; Alhadeff et al, 2019; Li et al, 2019). The subcellular mechanisms responsible for these changes are complex and are likely mediated by a number of intracellular signaling pathways. These are actively under investigation and are specifically being studied in AgRP neurons (Kong et al, 2016; Bruning & Fenselau, 2023; Grzelka et al, 2023). Moreover, how changes within AgRP neurons affect downstream neuronal networks and subsequent feeding behavior, be it food consumption or cessation of feeding, is an active area of research. Sayar-Atasoy and colleagues recently reported that AgRP neurons integrate information about circadian feeding with current metabolic needs to determine future feeding times (Sayar-Atasoy et al, 2024). This is relevant to studies of energy deprivation in rodents, as many studies use fasting paradigms from periods of 9 h up to 48 h (Wu et al, 2014; Steculorum et al, 2016; Gui et al, 2023), with the most widely used paradigm involving a 16-h fast with an onset at the beginning of the dark phase. Whereas an assessment of the effects of fasting (across different timespans) is known in peripheral tissues (liver and muscle) (Kinouchi et al, 2018), an assessment of the effects of fasting, across different Zeitgeber (ZT) timespans, on extra-hypothalamic neuronal circuits involved in the energy balance is lacking and was the primary goal of this study.

[1]Department of Neurocircuit Development and Function, German Institute of Human Nutrition, Nuthetal, Germany   [2]German Center for Diabetes Research (DZD), München-Neuherberg, Germany   [3]NeuroCure Cluster of Excellence, Charité-Universitätsmedizin, Berlin, Germany

Correspondence: rachel.lippert@dife.de
*Robert A Chesters and Jiajie Zhu contributed equally to this work

Studies of energy deprivation through fasting have assessed the expression of the immediate early gene, c-Fos, as a marker for increased neuronal activity and have shown increased levels in AgRP neurons within the ARC (Steculorum et al, 2016; Gui et al, 2023). However, studies assessing neuronal responses to fasting, in extra-hypothalamic brain regions are less prevalent, especially in regions known to have direct connectivity to the AgRP and POMC neurons of the ARC. Those that do exist have investigated the potential intersection of energy state and food intake, food choice and food reward (Zhang & van den Pol, 2017; Li et al, 2019; Mazzone et al, 2020).

The melanocortin 4 receptor has been the focus in studies of the hypothalamic AgRP and POMC neuronal projections (Atasoy et al, 2012), but it is not the only centrally expressed melanocortin receptor. The melanocortin 3 receptor (MC3R) is also expressed in a number of brain regions, independent of the melanocortin 4 receptor, suggestive of a different role for this centrally expressed melanocortin receptor (Roselli-Rehfuss et al, 1993). The MC3R acts as a central rheostat, coordinating not only aspects of energy excess but also energy deprivation (Ghamari-Langroudi et al, 2018). MC3R deficient animals show both a defective fasting response, highlighted by decreased refeeding upon food presentation, and changes to c-Fos activation in the ARC and in reward circuits (Renquist et al, 2012; Girardet et al, 2017; Ghamari-Langroudi et al, 2018; Gui et al, 2023).

Deficiency of the MC3R results in altered body composition, mild obesity, alterations to feeding behaviors such as food anticipatory activity, and changes to reward-related intake (Butler et al, 2000, 2017; Begriche et al, 2012; Lippert et al, 2014). Alterations in feeding behaviors have also been identified in humans with MC3R mutations (Obregon et al, 2010; Aris et al, 2016). Furthermore, recent studies have uncovered a role for the MC3R in a number of other, non-homeostatic aspects of energy balance and behavior, such as reward-based or stress-induced feeding (Pandit et al, 2015, 2016; Roseberry et al, 2015; Yen & Roseberry, 2015; Pei et al, 2019; West et al, 2019; Sweeney et al, 2021; Cho et al, 2023). Many of these actions have been targeted to extra-hypothalamic regions expressing the MC3R, highlighted by the recently updated neuroanatomical assessment of MC3R in the brain (Bedenbaugh et al, 2022).

In the context of extra-hypothalamic regions involved in food intake, a pivotal study assessing AgRP projection stimulation and food intake highlighted additional target regions, including the paraventricular nucleus of the thalamus (PVT) (Betley et al, 2013). Optogenetic activation of these AgRP neuronal projections to the PVT led to increased feeding (Betley et al, 2013). Moreover, AgRP neuronal projections to the PVT can facilitate food-seeking behaviors in the fasted condition (Wang et al, 2021). Recent interest has increased in probing how the PVT may interact with metabolic circuits (Kirouac, 2015; Millan et al, 2017; Petrovich, 2021). The PVT is a midline structure that receives inputs from multiple brain regions (Kirouac, 2015; Iglesias & Flagel, 2021), and is a relevant relay center involved in feeding behavior (Zhang & van den Pol, 2017; Cheng et al, 2018; Horio & Liberles, 2021), reward (Parsons et al, 2007; Clark et al, 2017; Do-Monte et al, 2017; Meffre et al, 2019; Otis et al, 2019; Iglesias & Flagel, 2021), fear (Li et al, 2014; Penzo et al, 2015), stress (Dong et al, 2020; Gao et al, 2020; Rowson & Pleil, 2021), and anxiety processing (Cho et al, 2023).

To identify PVT neurons mediating aspects of energy balance, we systematically characterized the neuroanatomical distribution of the MC3R within the thalamus and showed that it is highly expressed throughout the PVT. We also analyzed how this nucleus responds to two fasting conditions, overnight and daytime, by quantifying immediate early gene protein expression and phosphorylation of a known signaling mediator, which we hypothesized would be reduced within MC3R-positive neurons. Furthermore, we questioned whether perturbation of MC3R activity during fasting would result in changes to refeeding behaviors in a mouse model with Designer Receptors Exclusively Activated by Designer Drug (DREADD)-mediated activation or inhibition. This work shows the effects of time of day and energy state on PVT-MC3R neurons and highlights behaviors related to feeding which are mediated by MC3R activity in both male and female animals.

# Results

## MC3R robustly expressed in the thalamus, with predominant expression in the PVT

To investigate the expression of the MC3R in the thalamus, we used the MC3R-GFP mouse model where GFP is expressed under the direct control of the MC3R promoter. Neurons expressing MC3R were seen in the central medial nucleus (CM), the central lateral nucleus (CL), the intermediodorsal nucleus (IMD), and the PVT in both male and female mice (see Fig 1A–D). Quantification of individual neurons shows a predominant expression in the PVT (see Fig 1D). As the anterior to posterior axis of the PVT was recently shown to have functionally distinct cell types (Gao et al, 2020), we subsequently subdivided quantification of MC3R neurons across the anterior (aPVT), mid (mPVT), and posterior (pPVT) (see Fig 1A–C and E). We found the greatest number of MC3R neurons in the pPVT.

## Time of day influences the effect of fasting on PVT activation

After confirming the expression of the MC3R across the anterior to posterior axis of the PVT, we next sought to understand how neurons in the PVT respond to energy deprivation. Neuronal activity, and markers thereof, are used to assess steady-state chronic activation of neuronal circuits (Bartel et al, 1989; Sheng & Greenberg, 1990; Ginty et al, 1992; Ji & Rupp, 1997; Wu et al, 2001). Here we first assessed PVT activation in the context of fasting. In the field, a wide range of fasting times are used and up to 24–48 h of fasting has been shown to activate neuronal circuits (Wu et al, 2014; Girardet et al, 2017; Horio & Liberles, 2021; Cho et al, 2023; Gui et al, 2023). However, despite their tolerability, longer periods of fasting are associated with a physiological response more similar to starvation than normal physiological feeding (Moro & Magnan, 2021). Here we investigated two fasting periods, subjecting mice to either an overnight fast of 16 h (ZT12-ZT4) or a daytime fast (ZT4-ZT20, see Fig 2A), time locked to the peak night phase of normal food intake to replicate the previous c-Fos activation studies (Steculorum et al, 2016; Chen et al, 2023). In both time periods, fasted males and females lost weight, compared with their fed counterparts (see Figs 2B and S1B). Using the neuronal activity marker c-Fos, and with the exception of the pPVT in the male

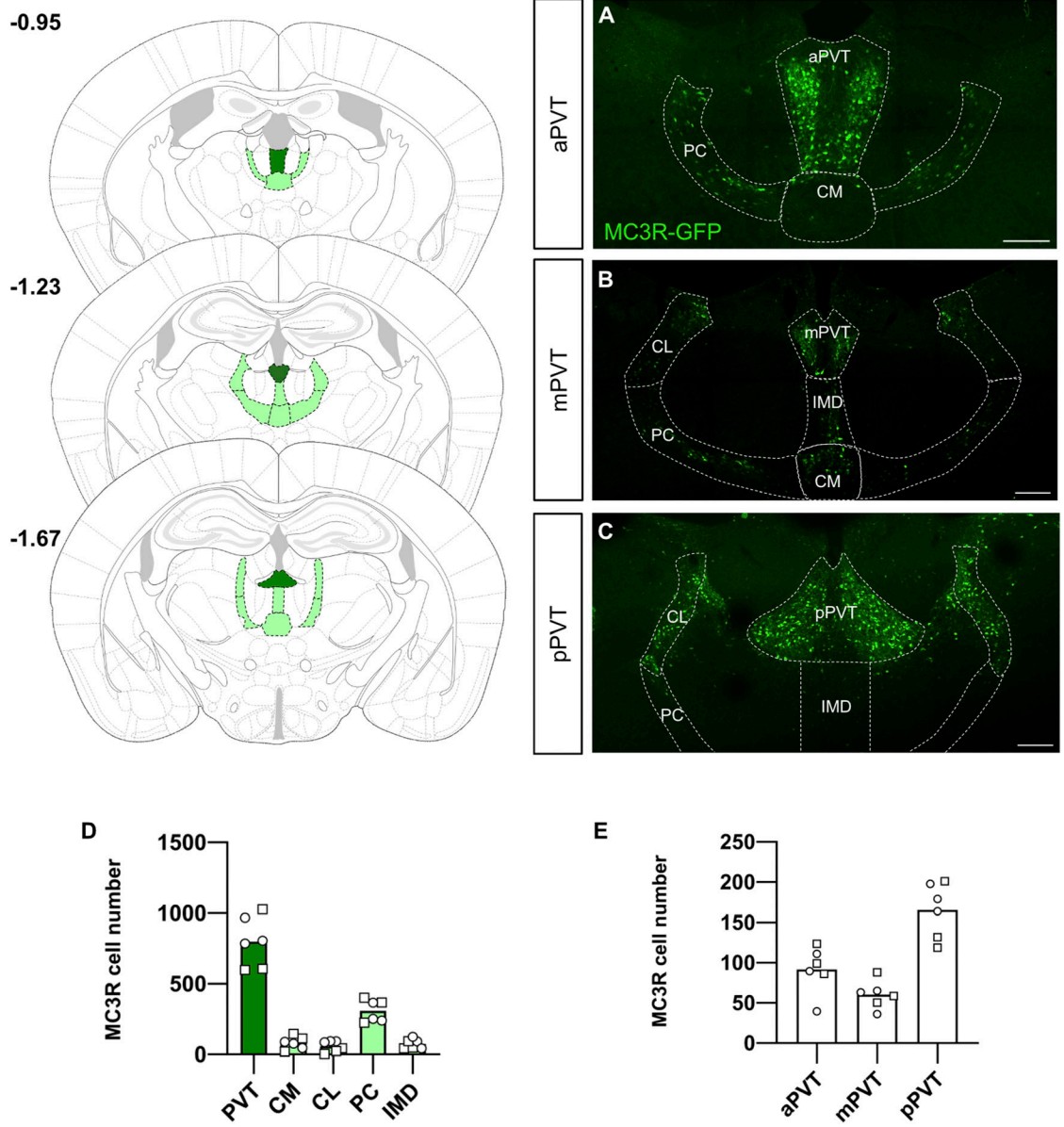

**Figure 1. Neuroanatomical distribution of melanocortin 3 receptor (MC3R) neurons in the thalamus.**
**(A, B, C)** Left panel shows representative images of thalamic nuclei at −0.95 (A), −1.23 (B), and −1.67 (C) relative to Bregma. **(D)** Quantification of MC3R neuronal number in the five most populated thalamic nuclei: paraventricular nucleus of the thalamus (PVT), central medial (CM), central lateral (CL), paracentral (PC), and intermediodorsal (IMD). **(E)** Quantification of MC3R neurons in different regions of the PVT: anterior (aPVT), mid (mPVT), and posterior (pPVT). Male data points are represented as circles and female data points as squares. Scale bars in (A, B, C) are 200 μm.
Source data are available for this figure.

daytime fasting experiment (Fig S1G–I, *P* = 0.014), in general, we did not see any differences across the anterior to posterior axis of the PVT between fed and fasted groups, neither in males or females, either in the daytime or overnight fast (see Figs 2C–P and S1C–P).

## pCREB cell density is decreased in response to fasting

Whereas c-Fos is a primary marker of neuronal activity, studies suggest that c-Fos expression in general is the result of multiple waves of neuronal activity and may not reflect more acute activation of neurons (Anisimova et al, 2023 *Preprint*). Thus, we opted to further probe upstream signaling pathways that would be activated upon stimulation of intracellular signaling cascades related to changes in energy state. We used immunolabeling of the phosphorylated form of the cyclic AMP response element binding protein (pCREB). CREB is an immediate upstream marker of c-Fos and was recently used to indicate significant changes to overall PVT activity (Kato et al, 2019). Analysis of pCREB immunolabeling showed a significant reduction in cellular density in the mPVT and pPVT in females in both the daytime (see Fig 3A–G) and the overnight (see

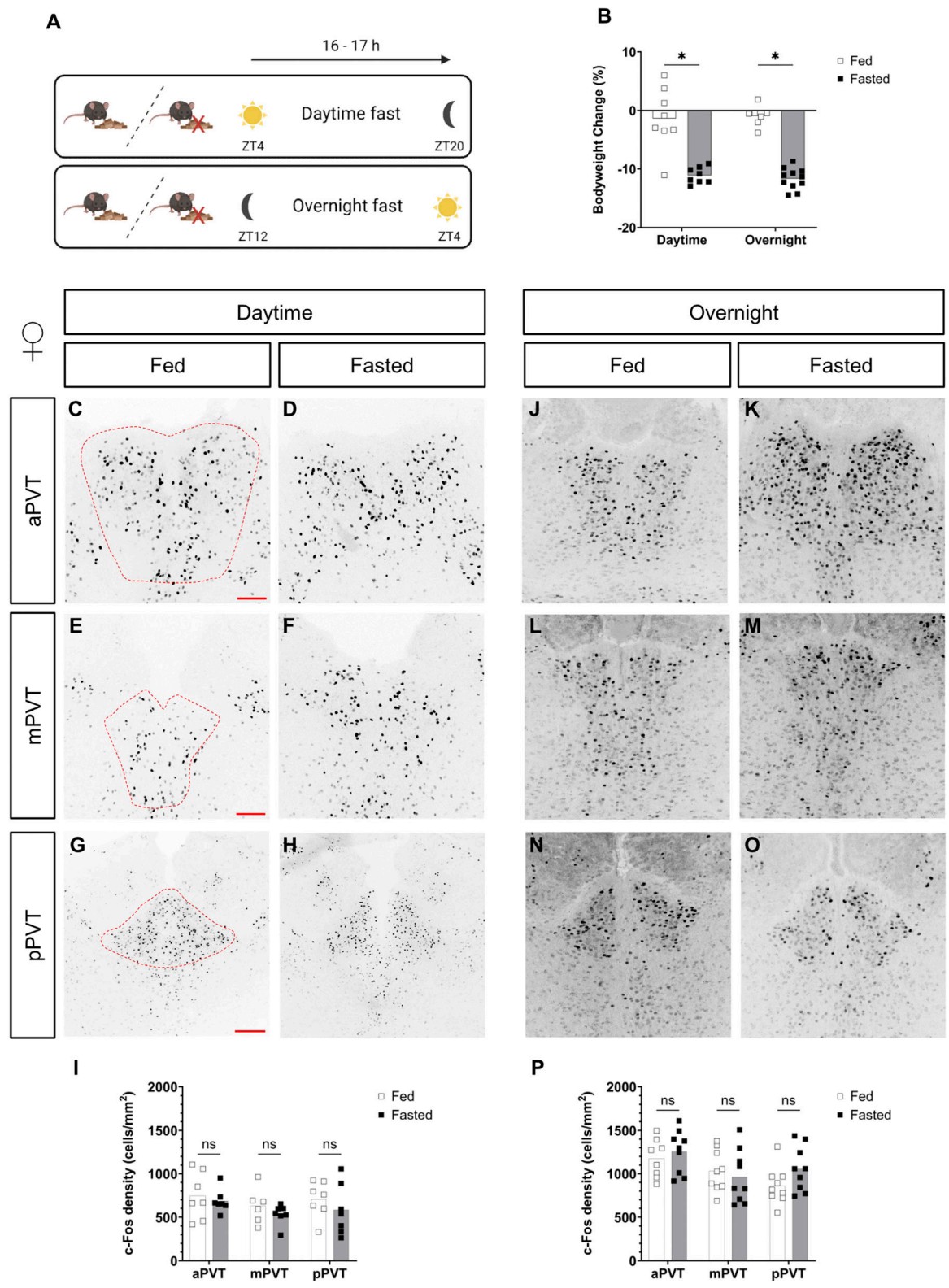

Figure 2. **Comparison of c-Fos cell density, in the paraventricular nucleus of the thalamus (PVT), of fed versus fasted female mice.**
**(A)** Experimental paradigm. Zeitgeber (ZT). **(B)** Percentage of body weight change at the start and end of the fasting period. **(C, D, E, F, G, H, I, J, K, L, M, N, O)** Representative images of c-Fos–positive cells in the anterior PVT (aPVT), mid PVT (mPVT), and posterior PVT (pPVT). **(C, E, G)** The red dashed outline in (C, E, G) represents regions of interest drawn around the PVT for analysis. **(I, P)** Quantification of c-Fos cell density in daytime (I) and overnight **(P)** fasting experiments. **(G)** Scale bars in (C, E) are 100 and 200 $\mu$m in (G). Unpaired Welch $t$ tests: *$P \leq 0.05$, ns, non-significant.
Source data are available for this figure.

Fig 3H–N) fasts. In males, there was no difference between groups, in the daytime fast (see Fig S2A–G). However, decreases in pCREB were seen in the overnight fast, and were significant in the pPVT (see Fig S2H–N). These changes indicate a more pronounced effect of fasting in females.

### MC3R neuronal activity markers decrease in the fasted state

To determine the direct effect of fasting on MC3R neuronal activity, colocalization of MC3R with c-Fos and pCREB immunolabeling was assessed using the MC3R-GFP mouse model. In line with the overall lack of effect on c-Fos activation in the PVT, and with the exception of the pPVT in daytime fasted female mice (Fig S3E–G, $P = 0.04$), in general, the colocalization of c-Fos with MC3R-GFP was no different between fed and fasted females (see Fig S3) or males (see Fig S4) in either the daytime or overnight fasts.

However, contrary to the lack of difference in the MC3R-c-Fos co-labeling, levels of MC3R-pCREB–positive neurons were significantly reduced in the PVT of females exposed to an overnight fast (see Fig 4H–N) but not in those exposed to a daytime fast (see Fig 4A–G). In males, there was no significant difference between groups, in MC3R-pCREB–positive cells, in either the daytime or overnight fasts (see Fig S5).

### AgRP neuropeptide levels and projections are not changed in the PVT in response to fasting

Recent studies suggest that in response to changes in energy state, the synaptic contacts or neuronal projections of AgRP neurons may be altered (Wei et al, 2015; Grzelka et al, 2023). To determine if changes in activity were linked to gross anatomical reductions in AgRP projection density, these were assessed in the PVT of fed and fasted, male and female mice. No group-level differences were observed in AgRP projections, in either male or female mice, in both the daytime and overnight fasts (see Figs S6 and S7). This suggests that changes to pCREB activation in the PVT are not because of significant alterations in AgRP projections to the region.

### Modulation of MC3R activity during fasting does not alter cumulative refeeding and body weight response

Reduced activity of MC3R neurons in response to a fast was shown at the level of pCREB cellular density in the PVT specifically in response to the overnight fasting paradigm. Here we aimed to determine how the direct activation state of MC3R cells may regulate the fasting response. Using a chemogenetic approach, we tested if heightened MC3R activity during the fasting period would result in suppression of body weight loss during fasting or in an alteration in refeeding. To address this question MC3R-Cre mice were crossed to floxed-hM3DGq mice to result in offspring with the activating DREADD receptor, hM3DGq, expressed in all MC3R cells in the body (see Fig S8). The fasting experiment (see Fig 5A) was performed with clozapine-*n*-oxide (CNO) application and subsequent activation of all MC3R cells. As a single injection of CNO can exert a biological effect up to 6–10 h, mice were injected twice across the dark period (Alexander et al, 2009; Wess et al, 2013; Biglari et al, 2021). No difference in body weight loss during fasting was detected in either females or males given CNO versus saline (see Figs 5 and S9). Furthermore, there was no difference in refeeding in the acute (1–12 h) or sustained (1–3 d) period after fasting (see Fig 5D and E). This was accompanied by no difference in the body weight regain during the refeeding period (see Fig 5C).

In a complementary experiment aiming to determine if MC3R inactivation during fasting may result in a reduced feeding re-bound after fasting, the inhibitory DREADD receptor, hM4DGi, was expressed in all MC3R-positive cells. As MC3R knockout animals are known to have a defective response to fasting (Renquist et al, 2012), we tested whether acute inhibition of MC3R cells during the fasting period would mimic the phenotype of the whole-body knockout animal to determine if the acute activity change in these neurons is necessary for refeeding responses. To do this, MC3R-Cre animals were crossed with floxed-hM4DGi animals, to result in hM4DGi expression localized to all MC3R-positive cells. Mice received acute injections of CNO at two timepoints across the fasting period to ensure sufficient inhibition (see Fig 6A). At the start of the light phase, mice were exposed to food, and their refeeding response was recorded. We did not find a difference in body weight loss because of fasting, in either fe-males or males given CNO versus saline (see Figs 6A and S10A). There were also no differences in food intake and body weight measurements, in CNO versus saline groups (see Figs 6B–D and S10B–D).

### Modulation of MC3R neurons during fasting changes behavioral patterns during initial refeeding

Although there were no differences in body weight or food intake across the hours and days post-fasting, we did see a significant difference in the behavior of the mice during the first hour of refeeding. We established an analysis pipeline using DeepLabCut (DLC) and Simple Behavior Analysis (SimBA) to assess the be-havior of the MC3R; hM3DGq and MC3R; hM4DGi mice (see Fig 7A and B). Specifically, we measured the time spent in different areas of the home cage, including time spent at the newly re-stocked food hopper. In MC3R; hM3DGq male and female animals (grouped together), CNO-treated mice spent significantly more time interacting with the food hopper (and the tip of the water bottle) than saline-treated mice (Fig 7C and D), whereas there was no significant difference between treatment groups in the MC3R and hM4DGi mice. Additional assessment of the time spent in the residual zone (Fig S11), consumption zone (food hopper + bottle tip + residual zone, Fig S12), total distance traveled (Fig S13), and velocity (Fig S14), did not show any significant difference between treatment groups. Thus, here we show that activation of the MC3R, but not inhibition, during the fasting period, can affect the subsequent behavior of mice during refeeding.

In summary, we have identified the expression of MC3R neurons in the thalamus, predominantly in the anterior to posterior axis of the PVT. Furthermore, we have uncovered that these neurons have a dynamic response to fasting dependent on the time of day of the fast, and that fasting-induced decreases in pCREB appear greater in female mice. We could demonstrate that MC3R modulation during the fasting

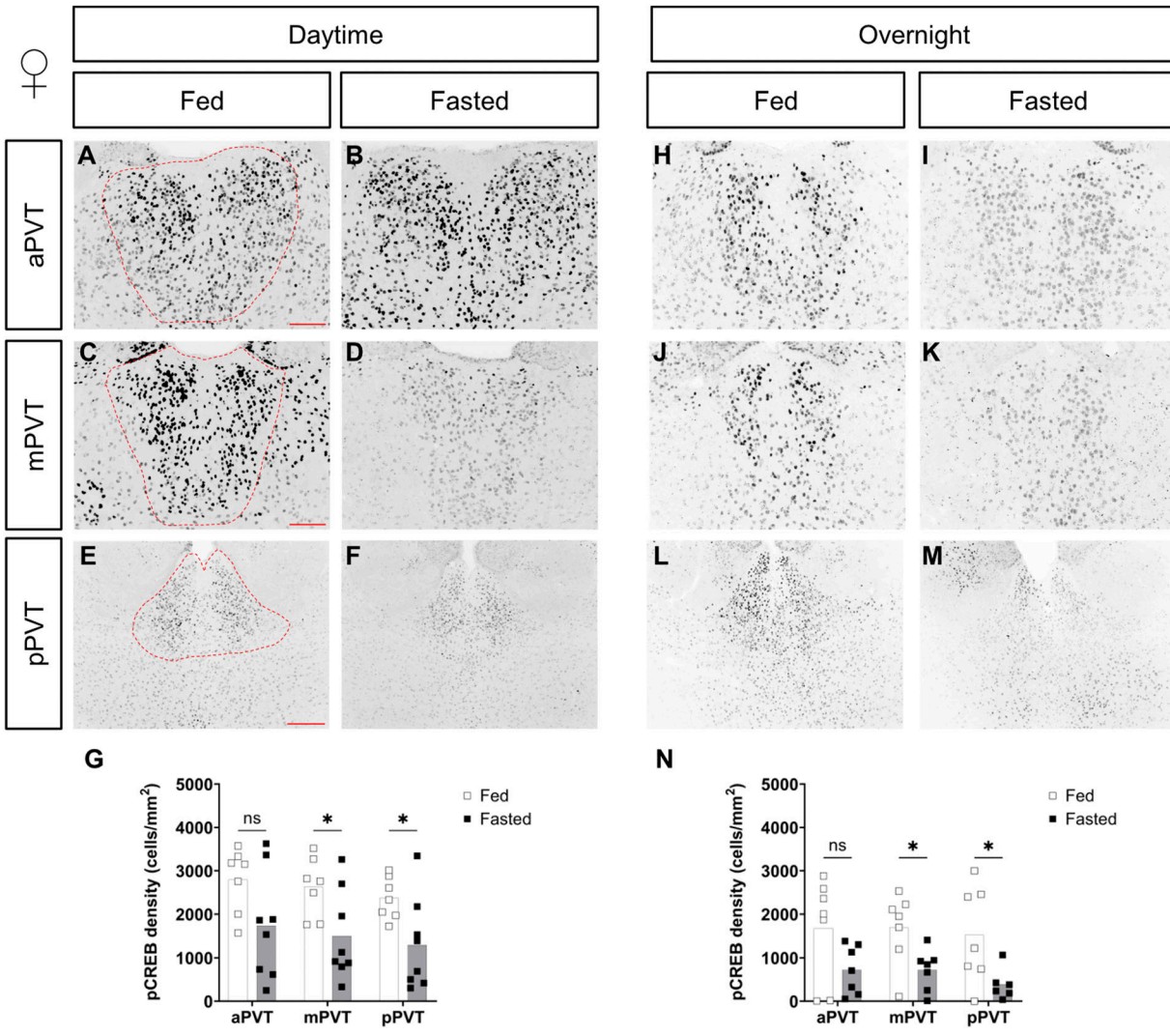

**Figure 3. Comparison of pCREB cell density, in the paraventricular nucleus of the thalamus (PVT), of fed versus fasted female mice.**
**(A, B, C, D, E, F, G, H, I, J, K, L, M)** Representative images of pCREB-positive cells in the anterior PVT (aPVT), mid PVT (mPVT), and posterior PVT (pPVT). **(A, C, E)** The red dashed outline in (A, C, E) represent regions of interest drawn around the PVT for analysis. **(G, N)** Quantification of pCREB cell density in daytime (G) and overnight (N) fasting experiments. **(E)** Scale bars in (A, C) are 100 and 200 $\mu m$ in (E). Unpaired Welch $t$ tests: *$P \leq 0.05$, ns, non-significant.
Source data are available for this figure.

period does not affect the amount of food eaten, during the refeeding period, nor the amount of body weight regain that occurs. However, we do show significant alterations in behavioral patterns in the immediate period after re-exposure to food. These results indicate that MC3R neuronal modulation may be necessary to fine tune the feeding behavioral response to result in a restoration of energy homeostasis.

## Discussion

Here we indicate distinct neuroanatomical distributions of the MC3R throughout various thalamic nuclei. Specific focus on the PVT shows extensive expression of the MC3R throughout the anterior to posterior axis of this nucleus. Probing the interaction of the PVT with energy state changes induced by fasting resulted in significant

overall decreases in pCREB activation in the PVT, with a specific effect within the MC3R neurons in this region. This alteration in activity was not associated with changes to AgRP neuropeptide projection density in the PVT. However, we cannot rule out changes to the overall neuronal activity of these AgRP neuronal projections, as fasting is known to increase neuronal activity within the ARC and increase the activity of AgRP neurons. Furthermore, optogenetic stimulation of AgRP projections to the PVT is known to increase feeding (Betley et al, 2013; Grzelka et al, 2023).

Inhibition of AgRP neuronal projections to PVT neurons in calorically restricted mice prevents food-seeking behavior (Wang et al, 2021). However, the opposite scenario, chronic activation of the circuit in an ad libitum state, does not elicit or drive food-seeking behavior (Wang et al, 2021). This suggests that AgRP to PVT connectivity is more strongly associated with seeking food in an energy-deprived state. Interestingly, whereas we did not see any

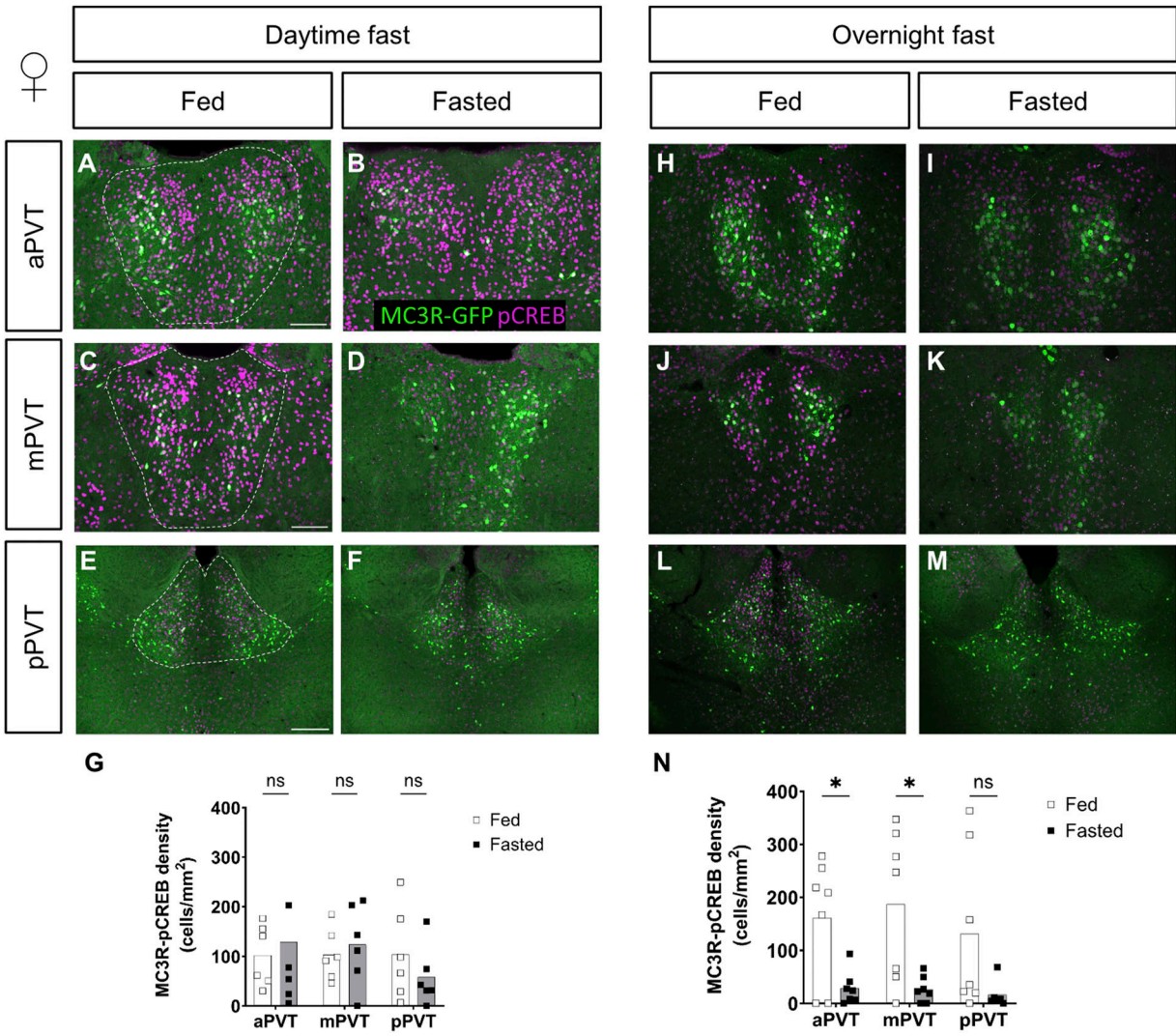

**Figure 4. Comparison of MC3R-GFP/pCREB colocalized cell density, in the paraventricular nucleus of the thalamus (PVT), of fed versus fasted female mice.**
**(A, B, C, D, E, F, G, H, I, J, K, L, M)** Representative images of MC3R-GFP– and pCREB-positive cells in the anterior PVT (aPVT), mid PVT (mPVT), and posterior PVT (pPVT). **(A, C, E)** The white dashed outline in (A, C, E) represent regions of interest drawn around the PVT for analysis. **(G, N)** Quantification of colocalized cell density in daytime (G) and overnight (N) fasting experiment. **(E)** Scale bars in (A, C) are 100 and 200 μm in (E). Unpaired Welch t tests: *P ≤ 0.05, ns, non-significant.
Source data are available for this figure.

significant alteration of body weight change upon refeeding, indicative of the matched food intake between the groups, an analysis of the time spent in various regions of the cage and distance traveled within the refeeding period uncovered unique changes. Activation of MC3R neurons during fasting led to greater time being spent at the food hopper (and the tip of the water bottle) after the food was present. This increased interaction with the food hopper is interesting given the documented role of MC3R in food anticipatory activity. In normal animals, an increase in locomotor activity is noted before the presentation of food; however, MC3R-deficient animals show a significant defect in food anticipatory activity (Sutton et al, 2008, 2010). Our data would suggest that hyperactivation of MC3R in the fasting period results in potential increases in food anticipatory behavior, food arousal or food attention. However, the exact behavioral changes and their underlying networks need further investigation. In our model, we cannot rule out that the chemogenetic modulation of neuronal activity is also affecting AgRP neurons, as up to 97% of AgRP neurons co-express the MC3R (Sweeney et al, 2021). Thus, when MC3R neurons, and by default many AgRP neurons of the ARC, are activated across the fasting period, this could potentially alter the neuronal response to food resulting in changes to feeding behavior. Indeed, one limitation of the model in our study is the expression of the DREADD receptors in all MC3R-positive cells throughout the body as driven by the MC3R-Cre model. However, as no difference in food intake itself was uncovered, it is likely that the MC3R effects are occurring outside the ARC. Subsequent studies will need to be performed to determine the possible contribution of peripheral MC3R activation and the specificity of PVT-MC3R cells in mediating this behavior.

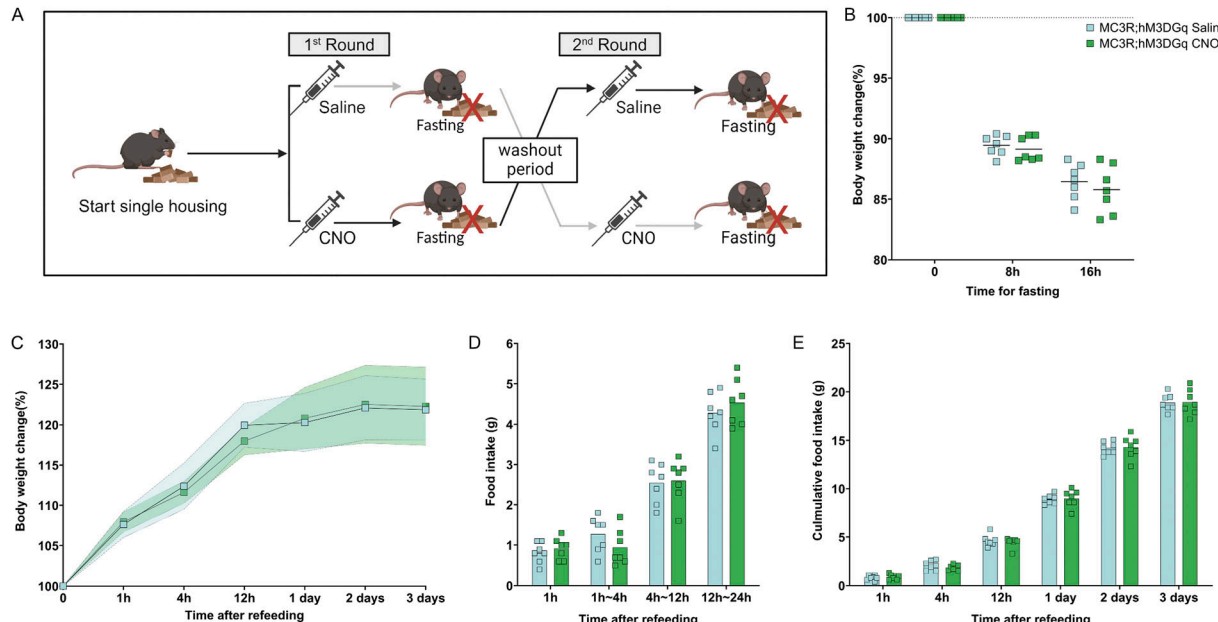

**Figure 5. Activation of melanocortin 3 receptor (MC3R) neurons during overnight fasting in females.**
**(A, B)** Experimental paradigm, (B) Body weight change during the fasting (in percentage) was plotted in comparison to the body weight at fasting, before the first injection. Body weight was measured at 8 h fasting before the second injection and 16 h before refeeding. **(C)** Body weight change after refeeding at different time points; data points indicate mean; colored area indicates SD. **(D)** Food intake was measured during refeeding at different time intervals. **(E)** Cumulative food intake (in grams) was measured during refeeding at different time points. **(B, C, D, E)** Blue indicates the MC3R; hM3DGq saline-treated control group and green indicates the MC3R; hM3DGq clozapine-n-oxide–treated group. N = 7 for both groups in all the graphs. Two-way ANOVA. No significance was detected.
Source data are available for this figure.

Recent heightened interest in the MC3R has proposed a functional role in driving sexually dimorphic anorectic behaviors. Specifically, probing PVT activity in response to refeeding after a fast shows dramatic increases in c-Fos expression across the anterior-to-posterior axis (Cho et al, 2023). Assessment of real-time activity of these neurons was probed in response to stressors in the environment and indeed the calcium-mediated activity was significantly altered in MC3R neurons. Here we show that, already in the fasting period, PVT neurons show decreased levels of pCREB in the mid and posterior PVT regions. Interestingly, this effect seems to be greater in female mice, which supports recent studies associating MC3R activity to anorexia-like phenotypes and increased effects on aspects such as novelty-suppressed feeding in female animals (Sweeney et al, 2021). In addition to this decrease in overall signal, there is also a significant overall decrease in pCREB labeling of the MC3R-positive neurons specifically. This was more dramatic in the overnight versus the daytime fasted groups, highlighting the likely importance of the recently described circadian rhythmicity of the AgRP neuronal activity in mediating aspects of this response (Sayar-Atasoy et al, 2024). In our study, the time of day of sample collection resulted in an apparent overall, and inverse, effect on c-Fos and pCREB labeling. This is consistent with known literature showing circadian rhythmicity of known hormones acting in this brain region, such as AgRP (Cedernaes et al, 2019; Sayar-Atasoy et al, 2024) or changes in glucocorticoids, whose circadian rhythm s also defective in MC3R KO animals (Renquist et al, 2012).

Whereas changes in markers of neuronal activity in MC3R neurons are noted and have sex-specific effects, the causative mechanism for this potential sexual dimorphism remains elusive. In addition, the exact downstream effects of changes to MC3R neuronal activity are not known. As more studies delineate the ARC[AgRP] to PVT[MC3R] neuronal connections and their role in various PVT-mediated behaviors, it will be critical to understand the real-time activity changes and modulation of further MC3R neuronal targets to build upon our knowledge of neurocircuit activity fluxes and effects on feeding behaviors in various metabolic and environmental contexts.

These changes to behavior and the underlying neuronal circuits in this acute (minutes to hours) time period are critical to understanding and improving interventional strategies for curbing altered feeding behaviors. These small alterations in feeding, the acute changes to neurocircuit activity and connectivity, and the associated changes to the whole-body physiological responses may ultimately result in a combined contribution to long-term metabolic dysfunction and the gradual accumulation of body weight.

# Materials and Methods

### Mice

All mice were bred within the animal facility at the Max-Rubner Laboratory at the German Institute for Human Nutrition, Potsdam-Rehbrücke (DIfE). Mice were group housed in individually vented

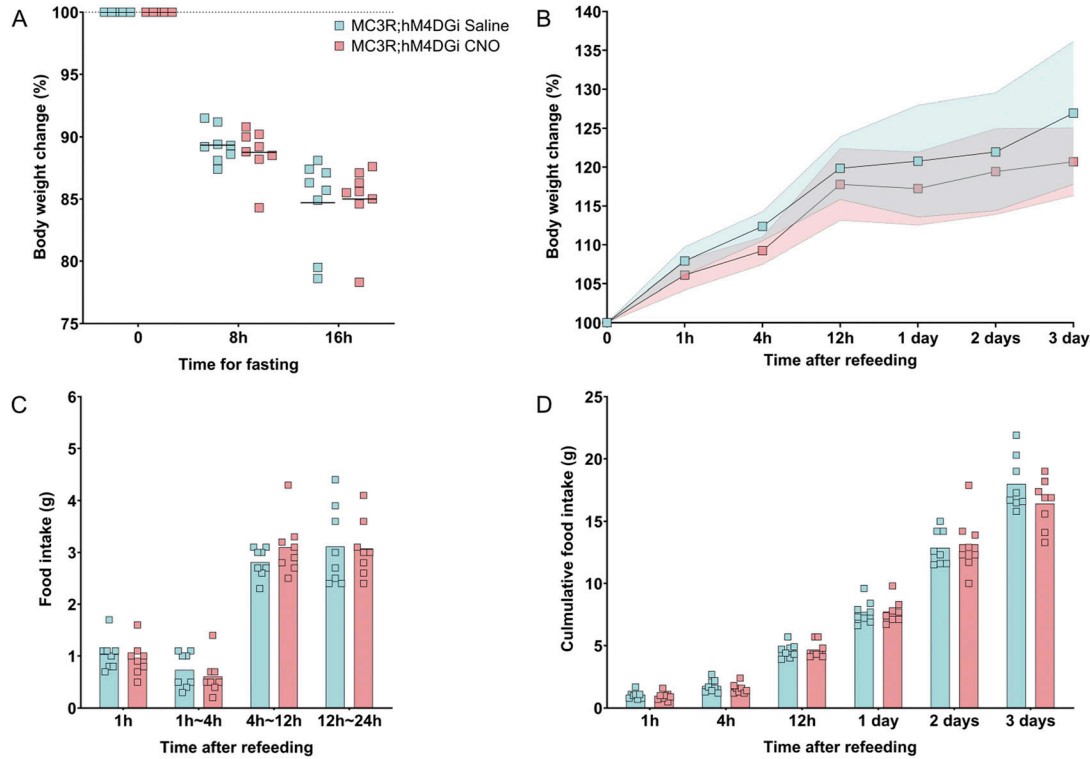

**Figure 6. Inhibition of melanocortin 3 receptor (MC3R) neurons during overnight fasting in females.**
**(A)** Body weight change during fasting (in percentage) was plotted in comparison to the body weight at fasting, before the first injection. Body weight was measured at 8 h fasting before the second injection, and 16 h before refeeding. **(B)** Body weight change after refeeding at different time points, data points indicate mean, colored area indicates SD. **(C)** Food intake (in grams) was measured during refeeding at different time intervals. **(D)** Cumulative food intake (in grams) was measured during refeeding at different time points. **(A, B, C, D)** Blue indicates MC3R; hM4DGi saline-treated control group and pink indicates MC3R; hM4DGi clozapine-*n*-oxide–treated group. N = 8 for both groups in all the graphs. Two-way ANOVA. No significance was detected.
Source data are available for this figure.

cages (IVC) cages, with ad libitum access (unless otherwise stated) to food and sterile acidified water, under a 12-h on/off light cycle, and an RT of 22 ± 2°C. All experiments were approved by the Landesamt für Arbeitsschutz, Verbraucherschutz, und Gesundheit (Land Brandenburg, Germany), under applications 2347-13-2021, 2347-42-2020, 2347-31-2021, and T-04-20-NDF, and conducted in compliance with the ARRIVE guidelines and the EU directive 2010/63/EU.

Transgenic MC3R-GFP (Tg [Mc3r-EGFP] BX153Gsat) (Gong et al, 2003; Heintz, 2004) mice were provided by Dr. Roger Cone and maintained on an FVB/N background. MC3R-Cre mice (Ghamari-Langroudi et al, 2018) were provided by Dr. Roger Cone and maintained on a C57BL6/N (>5 generations from C57BL6J) background. R26-LSL-RSR-hM4DGi-ZsGreen (De Solis et al, 2024) expressing mice that had previously bred to Deleter-Dre mice to remove the rox-flanked stop codon or R26-LSL-hM3DGq-eGFP (Steculorum et al, 2016) expressing animals were provided by Dr. Jens Brüning and were maintained on a C57BL6N background. When crossed with MC3R-Cre, this allows for the expression of the hM3DGq or hM4DGi in all MC3R-positive cells.

Genotyping of mice was carried out via PCR using the following primers and annealing temperatures listed in Table 1. The PCR cycling conditions were as follows: step 1: 95°C for 5 min; step 2:

95°C for 30 s; step 3: annealing (see Table 1); step 4: 72°C for 1 min (steps 2–4 repeated for 35 cycles); step 5: 72°C for 5 min. The resulting PCR products were separated on a 1% agarose gel.

### Fed-fasting

In all fasting experiments, 2–3-mo-old mice, in both fed and fasted groups, were placed into a new cage and access to food was removed from the fasted group for 16–17 h. Overnight fasting began at "Lights OFF" (ZT12). Daytime fasting began 4 h after "Lights ON" (ZT4). In Figs 2, 3, 4, S1, S2, S3, S4, S5, S6, and S7, mice in both fed and fasted groups were euthanized at the end of the fasting period.

### Fasting refeed

The fasting refeed experiment (in Figs 5 and 6) used 2–3-mo-old MC3R-Cre; ROSA26-LSL-hM3DGq-eGFP and MC3R-Cre; ROSA26-LSL-hM4DGi mice. For more accurate measurement of food and avoidance of fighting during fasting, animals were single housed. They were allowed to acclimate to single housing and human handling for 1 wk. Intraperitoneal (i.p.) injections of saline were given for 3 d before fasting to minimize the stress caused by injection. Upon fasting, the mice were weighed and then received i.p. injections of either CNO (MC3R-Cre; ROSA26-LSL-hM3DGq mice, 0.3

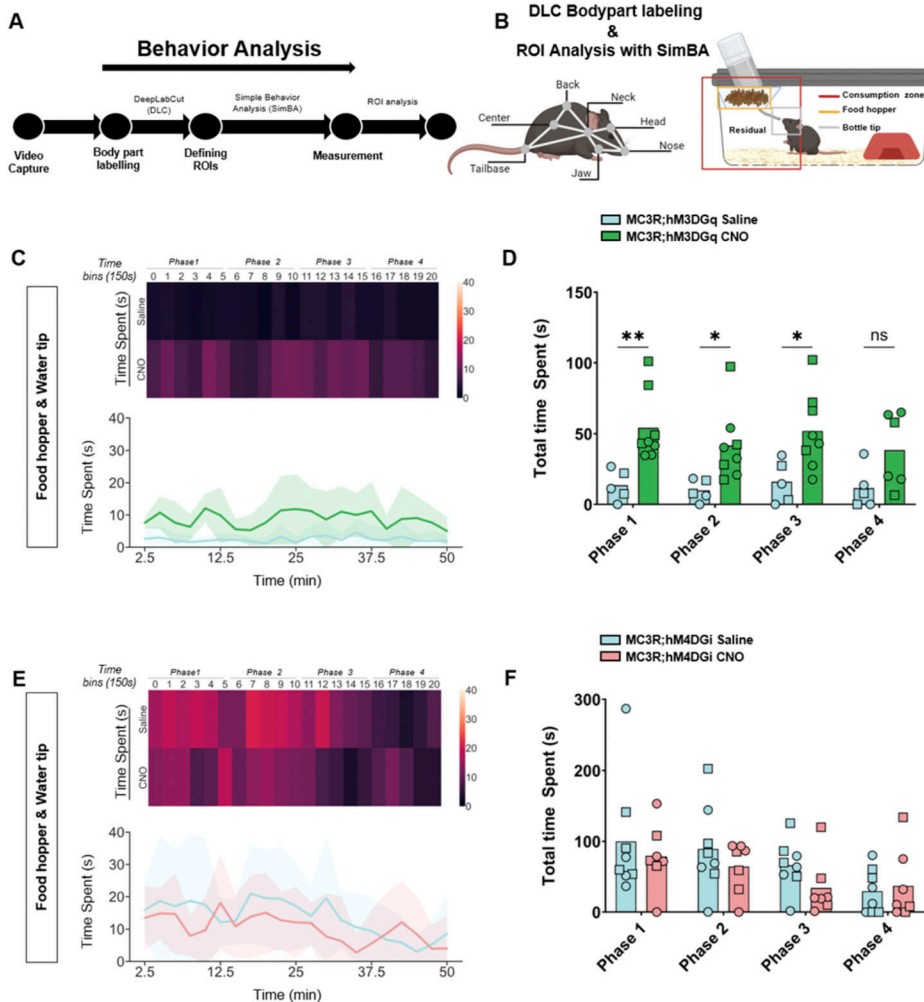

**Figure 7. Behavior tracking during refeeding after melanocortin 3 receptor (MC3R) activation or inhibition.**
**(A)** Analysis pipeline after video capture of animal behavior with refeeding. **(B)** Indications of body part labeling with DeepLabCut and analysis of regions of interest with SimBA. Selection of regions of interests within the home cage is indicated in the graphic. **(C)** Time spent at the food hopper (and bottle tip) in saline- and clozapine-*n*-oxide (CNO)–treated MC3R; hM3DGq mice visualized with a heat map ((C) top) or line graphs ((C) bottom). **(D)** Total time spent at the food hopper (and bottle tip) during time bins 0–5 (Phase 1), 6–10 (Phase 2), 11–15 (Phase 3), and 16–20 (Phase 4). Blue indicates the MC3R; hM3DGq saline-treated control group and green indicates the MC3R; hM3DGq CNO-treated group. Male data points are represented as circles and female data points as squares. **(E)** Time spent at the food hopper (and bottle tip) in saline- and CNO-treated MC3R; hM4DGi mice visualized with a heat map ((E) top) or line graphs ((E) bottom). **(F)** Total time spent at the food hopper (and bottle tip) during time bins 0–5 (Phase 1), 6–10 (Phase 2), 11–15 (Phase 3), and 16–20 (Phase 4). Blue indicates the MC3R; hM3DGq saline-treated control group and pink indicates the MC3R; hM4DGi CNO-treated group. Male data points are represented as circles and female data points as squares. **(D)** Two-way ANOVA (D, F), with Šídák's multiple comparisons test (D): *$P \leq 0.05$, **$P \leq 0.01$, ns, non-significant.
Source data are available for this figure.

**Table 1. Genotyping primers and annealing temperatures.**

| | Target | Sequence (5′-3′) | Product | Annealing temperature (time) |
|---|---|---|---|---|
| 1 | MC3R-Cre | Cre_WT (mom0137) CTGCAGTTCGATCACTGGAAC | WT: 542 bp | 66°C (30 s) |
| | | Cre_Common (mom0138) AAAGGCCTCTACAGTCTATAG | | |
| | | Cre_Mut1 (oIMR6694) TCCAATTTACTGACCGTACA | Tg: 450 bp | |
| | | Cre_Mut2 (oIMR6695) TCCTGGCAGCGATCGCTATT | | |
| 2 | MC3R-GFP | GFP116 Rev: ACA GCT CGT CCA TGC CGA GAG | WT: 325 bp | 56°C (45 s) |
| | | GFP115 Fwd: GAT GCC ACC TAC GGC AAG CTG | | |
| | | MC3R_GFP WT rev: GATGAAGACCTGCTCACAGAACCCAC | Tg: 600 bp | |
| | | MC3R_GFP Fwd: AGGAAAGTTCTTTCTATGTCTCCAAGCCC | | |
| 3 | hM3DGq and hM4DGI | Forward CAGGS: AAA GTC GCT CTG AGT TGT TAT C | WT: 570 bp | 56°C (45 s) |
| | | Rev-WT: GAT ATG AAG TAC TGG GCT CTT | | |
| | | Rev CAGGS: TGT CGC AAA TTA ACT GTG AAT C | Floxed: 380 bp | |

mg/kg, 10 µl/g BW and for MC3R-Cre; ROSA26-LSL-hM4DGi 3 mg/, 10 µl/g BW) or saline (0.9%, 10 µl/g BW). The same solutions were then given again 8 h later, to ensure activation/inhibition of MC3R neurons throughout the fasting period. Mice were fasted overnight, from ZT 11 to ZT 3. At refeeding, the mice were video recorded. Their body weight change and food intake were measured at 1, 3, 6, 12, and

**Table 2.  Primary antibodies used in immunohistochemistry experiments.**

| | Primary antibody | Host | Supplier | Catalog number | Dilution |
|---|---|---|---|---|---|
| 1 | Agouti-related peptide (AGRP) | Rabbit | Phoenix | H-003-53 | 1 in 40,000 |
| 2 | c-Fos | Rat | Synaptic Systems | 226 017 | 1 in 1,000 |
| 3 | GFP | Rabbit | Invitrogen | A-6455 | 1 in 5,000 |
| 4 | GFP | Chicken | Abcam | ab13970 | 1 in 5,000 |
| 5 | Phospho-CREB (ser133) | Rabbit | Cell Signaling | #9198 | 1 in 2,000 |

**Table 3.  Secondary antibodies used in immunohistochemistry experiments.**

| Secondary antibody | Supplier | Catalog number | Dilution | Used against primary antibody |
|---|---|---|---|---|
| Alexa Fluor 594 donkey anti-rabbit | Invitrogen | A21207 | 1 in 500 | 1 and 5 |
| Alexa Fluor 633 donkey anti-rat | Sigma-Aldrich | SAB4600133 | 1 in 500 | 2 |
| Alexa Fluor 488 donkey anti-rabbit | Invitrogen | A21206 | 1 in 500 | 3 |
| Alexa Fluor 488 donkey anti-chicken | Dianova | 703-545-155 | 1 in 1,000 | 4 |

24 h time points. To minimize animal numbers required for this experiment, a cross-over design was implemented: after a 2-wk wash-out period, the same mice underwent the same treatment procedure, but with the opposite treatment option (CNO or saline).

### Euthanasia and tissue processing

At euthanasia, mice were transcardially perfused with PBS followed by 4% PFA in borate buffer (pH 9.5) (#441244; Sigma-Aldrich), under pentobarbital anaesthesia (400 mg/kg; i.p.) diluted in isotonic sodium chloride solution (#1021010; Delta-medica). Whole brains were harvested and post-fixed in 4% PFA for 4–24 h, at 4°C. Brains were then dehydrated in 20% sucrose in PBS, overnight at 4°C, before being frozen at –80°C for further processing.

Frozen brains were then sliced into 30μm-thick coronal sections using a sliding microtome (#400410, Slide 4004 M; pfm medical) and serial sections were stored in glycerol-containing PBS solution at –20°C, until further use.

### Immunohistochemistry

Coronal brain sections including the PVT were first treated in 0.3% glycine for 10 min, followed by permeabilization for 10 min in 0.03% SDS. Next sections were blocked in 3% normal donkey serum in 0.125% Triton-X in K-PBS for 1 h at RT. Slices were then immunostained with primary antibodies in SignalStain solution (#8112; Cell Signaling) at 4°C overnight, as detailed in Table 2. Next, sections were washed in K-PBS 3 times for 10 min before being labeled with secondary antibodies, as detailed in Table 3. Finally, stained sections were mounted on Superfrost glass slides (J800AMNZ; Epredia) and coverslipped with VECTASHIELD Antifade Mounting Medium containing DAPI (#VEC-H-1200; Biozol).

### Image acquisition and analysis

Multi-channel Z-stack images of representative sections of the anterior PVT (from bregma –0.71 to –0.95 mm), mid PVT (from bregma –1.23 to –1.43 mm), and posterior PVT (from bregma –1.67 to –1.79 mm) were acquired at 20× magnification, using either a Zeiss Confocal Microscope (GFP-c-Fos and GFP-AgRP images) or a Zeiss Fluorescent Axio Observer Microscope equipped with Apotome 3 (GFP-pCREB images). One image per PVT region was acquired. Image processing and analysis were performed using custom-written macros in ImageJ software v2.1.0 (National Institute of Health).

For all image analysis, multi-channel z-stack images were compressed into a single plane in the z-axis using the maximum intensity projection function and split into separate channels. A region of interest (ROI) was manually drawn around the limits of the PVT using the GFP channel from each image and applied to the channel to be analyzed (GFP/c-Fos/pCREB/AgRP).

For cell density measurements, a threshold was automatically determined for the channel to be analyzed and converted to a binary image. In the GFP-c-Fos staining, the c-Fos channel was thresholded using the *Triangle* algorithm, and the GFP channel using the *Moments* algorithm. In the GFP-pCREB staining, a threshold for both channels was determined using *Triangle*. Images were then *despeckled* to remove noise and *watershed* to define individual cells. The *Analyze particles* plugin was then used, with the following minimum cell size, to automatically count the number of positive immunolabeled cells in the ROI: c-Fos and pCREB = 20 $\mu m^2$, GFP and colocalized cells = 40 $\mu m^2$. The number of cells was then divided by the ROI area to yield cell density measurements.

For fiber density measurements, the AgRP channel was automatically thresholded using the *Triangle* algorithm in ImageJ and converted to a binary image. The raw integrated density value (representing the total number of pixels containing positive immunolabeling) was measured and divided by the total area of the

ROI analyzed, yielding the percentage area of positive immuno-labeling (fiber density).

For both cell density and fiber density measurements, thresholded binary images were compared with the original images as a quality check. Images whereby the thresholding did not accurately detect cells/axon fibers were excluded from further analysis.

### DeepLabCut model and SimBA ROI analysis of refeeding behavior

During the refeeding experiment, Anymaze software (Stoelting) was used to generate video recordings of mice in regular IVC cages after a 16-h fast. Two machine-learning toolkits for animal behavior analysis were used, DeepLabCut (DLC) (Mathis et al, 2018) and Simple Behavior Analysis (SimBA) (Popik et al, 2023). These software programs were downloaded from (https://github.com/DeepLabCut) and (https://github.com/sgoldenlab/simba), respectively, and installed on a workstation with the following specifications: AMD Ryzon 9 7900X 12-Core Processor, with a speed of 4.70 GHz and RAM of 64.0 GB in Windows 10. Using the DLC version 2.3.5, a 7-body part labeling system was implemented, namely the nose, head, jaw, neck, back, center, and tail base (see Fig 7B). An experimenter blinded to the experiment manually labeled a total of 300 randomly sampled frames from 10 videos. The labeled frames were then used to train a machine model of 700,000 epochs or iterations with a $p_{cutoff}$ of 0.6. Validation of the trained network yielded the following results for 700,000 iterations: a training error of 1.12 pixels and a test error of 2.7 pixels. The trained model was then validated on a separate set of videos and finally used to analyze all video recordings. The subsequent DLC pose estimation coordinates of all labeled body parts were then transferred into SimBA. In SimBA, a novel 7-node-body configuration was generated based on the captured videos, which were side views, for classical tracking of mice. Using this side-view tracking format, the DeepPoseKit CSV files were imported from DLC, and the following SimBA modifications were carried out. First, the imported CSV files were interpolated using the quadratic interpolation method together with Gaussian smoothing of 200 ms. Afterward, a distance of 195 mm was used to calibrate the pixels to the distance for each video, which is the height of the IVC cages used in the experiments. Furthermore, an outlier correction with 1.0 criteria was performed for both location and movement, respectively, together with the mean aggregation method. This step was critical to excluding all non-specific labeling. Then, 3 ROIs were drawn, namely, the water, food, and consumption zones, as shown in Fig 7B. The distance traveled (m) and velocity (cm/s) of each mouse were measured using the ROI Data Aggregates and time bins analysis functions in SimBA. The time spent within each of the ROIs was quantified using the probability threshold of 0.25, to account for instances where the mouse enters the provided housing structure and thus may be absent from the frame. The analysis of the videos was split into time bins of 150 s for a total of 20-time bins corresponding to 50 min of behavior video capture. The read-outs from SimBA were further processed using Python (3.11.4) to measure time spent, distance traveled, and velocity: The time spent by each mouse in the residual zone was estimated by subtracting the time spent at both the food hopper and bottle tip from the time spent at the consumption zone.

### Statistical analysis

Bar graphs show all individual data points and group means. All graphs and inferential statistics were created using GraphPad Prism 10.1.2 (GraphPad Software) or performed in Python. Unpaired two-tailed Welch's $t$ tests were performed to analyze cell density and fiber density differences between groups. For behavioral tracking data across time, a two-way ANOVA was performed to determine overall main effects, and followed up (where necessary) with Šídák's multiple comparisons test. Statistical significance was set at an alpha value of 0.05. All data pertaining to specific statistical tests, n-numbers and significance can be found summarized in Table S1.

## Supplementary Information

## Acknowledgements

We would like to acknowledge all members of the Max-Rubner Laboratory, specifically Anja, Julia, Jasmin, Annelie, Martina, and Johanna for all of their support in the maintenance and controlling of the animals during the studies. We would also like to thank Dr. Roger Cone for providing the MC3R-GFP and MC3R-Cre mice and Dr. Jens Brüning for the provision of the ROSA26-LSL-hM3D$_{Gq}$ and ROSA26-LSL-RSR-hM4D$_{Gi}$ mouse models. We would like to thank Hamid Taghipourbibalan for his initial support in establishing various analysis pipelines for the assessment of behavior. Schematic diagrams of experimental design were created with BioRender.com. Funding support for the project was provided by the Leibniz Association through the Leibniz Competition Best Minds Grant "BAByMIND" (J99/2020) to RN Lippert, the Deutsche Forschungsgemeinschaft (DFG, German Research Foundation) under Germany's Excellence Strategy – EXC-2049 – 390688087 (NeuroCure, to RN Lippert), and by the German Center for Diabetes Research (82DZD03D2Y and 82DZD03D03 to RN Lippert). Open Access publication support was provided by the DFG (491394008).

### Author Contributions

RA Chesters: conceptualization, data curation, software, formal analysis, supervision, validation, investigation, visualization, methodology, project administration, and writing—original draft, review, and editing.
J Zhu: conceptualization, data curation, formal analysis, supervision, validation, investigation, visualization, methodology, project administration, and writing—original draft, review, and editing.
BM Coull: conceptualization, data curation, software, formal analysis, supervision, validation, investigation, methodology, project administration, and writing—review and editing.
D Baidoe-Ansah: data curation, software, formal analysis, supervision, validation, visualization, methodology, and writing—review and editing.
L Baumer: data curation, formal analysis, investigation, and writing—review and editing.
L Palm: data curation, formal analysis, investigation, and writing—review and editing.

N Klinghammer: data curation, formal analysis, and writing—review and editing.

S Chen: investigation and writing—review and editing.

A Hahm: investigation and writing—review and editing.

S Yagoub: investigation and writing—review and editing.

L Cantacorps: investigation and writing—review and editing.

D Bernardi: investigation and writing—review and editing.

K Ritter: investigation and writing—review and editing.

RN Lippert: conceptualization, data curation, formal analysis, supervision, funding acquisition, validation, investigation, visualization, methodology, project administration, and writing—original draft, review, and editing.

## Conflict of Interest Statement

The authors declare that they have no conflict of interest.

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
