## [Reviewer comments · Life Science Alliance]

Life Science Alliance

Fasting-induced activity changes in MC3R neurons of the paraventricular nucleus of the thalamus

Robert Chesters, Jiajie Zhu, Bethany Coull, David Baidoe-Ansah, Lea Baumer, Lydia Palm, Niklas Klinghammer, Seve Chen, Anneke Hahm, Selma Yagoub, Lídia Cantacorps, Daniel Bernardi, Katrin Ritter, and Rachel Lippert

DOI: <https://doi.org/10.26508/lsa.202402754>

Corresponding author(s): *Rachel Lippert, German Institute of Human Nutrition*

Review Timeline:	Submission Date:	2024-04-04
	Editorial Decision:	2024-05-13
	Revision Received:	2024-07-22
	Editorial Decision:	2024-07-23
	Revision Received:	2024-07-26
	Accepted:	2024-07-29

Transaction Report:

May 13, 2024

Re: Life Science Alliance manuscript #LSA-2024-02754-T

Dr. Rachel Nicole Lippert
German Institute of Human Nutrition in Potsdam (DIfE)
Junior Research Group Neurocircuit Development and Function
Arthur-Scheunert-Allee 114-116
Nuthetal 14558
Germany

Dear Dr. Lippert,

Thank you for submitting your manuscript entitled "Fasting induced changes in the PVT and the melancortin-3-receptor (MC3R)" to Life Science Alliance. The manuscript was assessed by expert reviewers, whose comments are appended to this letter. We invite you to submit a revised manuscript addressing the Reviewer comments.

Thank you for this interesting contribution to Life Science Alliance. We are looking forward to receiving your revised manuscript.

Sincerely,

B. MANUSCRIPT ORGANIZATION AND FORMATTING:

Reviewer #1 (Comments to the Authors (Required)):

This manuscript by Chesters et al., characterizes the neuroanatomical distribution of MC3R in the PVT and how this nucleus is sensitive to energy state using IEG protein and phosphorylation quantification. They then use chemogenetic manipulations to stimulate or silence MC3R-expressing PVT neurons to investigate feeding patterns. Overall, this is a very thorough study that does a good job systematically measuring the activity alterations in a specific subset of MC3R positive cells.

Do the authors know why there seems to be higher Fos levels during the overnight vs daytime measurements, independent of fed versus fasted? And same for pCREB, just vice versa. Is there some circadian control of these cells?

There are a couple of outliers in the pCREB quantification data that have a 0 cells/mm². Are those animals in which the staining didn't work, in other words did the authors look in other areas of the brain to ensure proper detection of pCREB was observed?

Was the percent change in bodyweight loss comparable in females versus males? The PVT is also a known stress center so it would be insightful to know if these animals were individually housed, acclimated, ect.

I would make it abundantly clear that the DREADD experiments were performed in all MC3R-expressing cells, beyond just those in the PVT. Although they don't claim they are just manipulating PVT MC3R neurons, it can be interpreted that way based on the flow of the manuscript.

Related, a major question remains what would be the effect on feeding/body weight if manipulations were specific to MC3R expressing PVT neurons.

It would be good to show some demonstration that CNO activates (hm3dq) or inhibits (hm4di) MC3R cells either via Fos staining or electrophysiology. Images of the expression of these DREADDs should also be shown.

Do the authors have a hypothesis on why acute modulation of MC3R neurons changes the amount of time interacting with the food hopper? Are they not eating more during Phase 1? Is this related to food anticipatory behavior or general arousal? Can these be tested to infer what these results may mean?

I'd recommend shorting the Introduction.

Reviewer #2 (Comments to the Authors (Required)):

This study examined the impact on MC3R expressing neuron activity by fasting during day or night phased and then the impact of increase or reduction in MC3R neurons on behaviors including feeding or drinking. Although analyses and individual experimental designs are sound, this manuscript is difficult to follow without a core hypothesis or a key insight revealed.

1) This study suffers greatly from lacking a key question to be addressed. This reviewer is confused why the study was conducted. The goal is to study MC3R or MC3R neuron function? or to study circadian regulation of MC3Rs? Many components seem to be involved but there appears to be no coherency here.

2) There seems to be a big gap between anatomical studies (c-Fos and pCREB) and DREADD studies. The former is on MC3R neurons in PVT; however, the latter is all MC3R neurons/cells in the whole body. Therefore it is difficult to integrate the results from these studies.

We thank the reviewers and the editor for reading the manuscript, the critical comments and the opportunity to revise the submission. We have aimed to address the comments and questions raised by the reviewers and we feel that the manuscript is now more coherent and is a better overall representation of the studies performed. Thank you for the feedback. We have addressed the reviewer comments as follows:

Reviewer #1 (Comments to the Authors (Required)):

1) This manuscript by Chesters et al., characterizes the neuroanatomical distribution of MC3R in the PVT and how this nucleus is sensitive to energy state using IEG protein and phosphorylation quantification. They then use chemogenetic manipulations to stimulate or silence MC3R-expressing PVT neurons to investigate feeding patterns. Overall, this is a very thorough study that does a good job systematically measuring the activity alterations in a specific subset of MC3R positive cells.

Thank you to the reviewer for the precise summary of the study performed.

2) Do the authors know why there seems to be higher Fos levels during the overnight vs daytime measurements, independent of fed versus fasted? And same for pCREB, just vice versa. Is there some circadian control of these cells?

This is a great observation and one that we have considered. Given the recent studies by Sayar-Atasoy et al and Cedernaes et al showing circadian rhythmicity of AgRP neurons, as well as the known circadian rhythm of hormones acting in this region (e.g. glucocorticoids, whose rhythm is also dysfunctional in MC3R KO animals – Renquist et al PNAS 2012), it is highly likely that these cells function under a circadian rhythm. To reinforce this point we have extended the discussion to point out the overall fluctuation that appears in the samples and how this may relate to the known findings in the literature.

Discussion Lines 455-458

“ (18). In our study the time of day of sample collection resulted in an apparent overall, and inverse, effect on c-Fos and pCREB labeling. This is consistent with known literature showing circadian rhythmicity of known hormones acting in this brain region, such as AgRP ((18, 77) or changes in glucocorticoids, whose circadian rhythm interestingly is also defective in MC3R KO animals ((27)).”

3) There are a couple of outliers in the pCREB quantification data that have a 0 cells/mm². Are those animals in which the staining didn't work, in other words did the authors look in other areas of the brain to ensure proper detection of pCREB was observed?

In all of our images, we performed a quality control to assess the overall automated detection of cells. In these particular images, none of the labeled cells passed the threshold required to be counted. Moreover, we include for the reviewer, images from other regions of the same brain slices where pCREB signal is detected. In these images pCREB positive cells are detected in the hypothalamus, but not in the PVT. Thus, we can conclude that the pCREB labelling was successful and have confidence in the results that we present.

Figure 1: Comparison of pCREB staining in the paraventricular thalamus (PVT) and ventromedial hypothalamus (VMH). A) x10 magnification image of a coronal brain slice at bregma -1.67. B-E) x20 magnification images of MC3R-GFP (C), pCREB (D) and DAPI (E) staining in the PVT. F-I) x20 magnification images of MC3R-GFP (G), pCREB (H) and DAPI (I) staining in the VMH. pCREB positive cells are clearly identifiable in the VMH (H), but not the PVT (D).

4) Was the percent change in bodyweight loss comparable in females versus males? The PVT is also a known stress center so it would be insightful to know if these animals were individually housed, acclimated, ect.

Thank you for this point. We have elaborated more on the experimental timeline in the methods section to point out the acclimatization steps taken in the fasting studies. For the DREADD studies, all animals were individually housed one week prior to the study and acclimated to injections for 3 days prior to start. For cohesiveness of graphs and to respond to the reviewers point, we have now plotted all data in Figures 5 and 6 as well as Supplemental Figures 8 and 9 as a percentage body weight lost.

The following text can be found in the methods section:

“For more accurate measurement of food and avoidance of fighting during fasting, animals were single housed. They were allowed to acclimate to single housing, as well as human handling for one week. Intraperitoneal (i.p.) injections of saline were given for three days before fasting to minimize the stress caused by injection.”

For the comparison of the percent change in body weight of males and females, we generated the graph below for the reviewer. In the hM3D experiment there is no detectable sex difference using a Two-Way ANOVA. With the hM4D line there is a small but significant sex difference, however this seems to be driven by the three animals that had significantly more weight loss. Therefore, we have opted not to include this data in the manuscript especially considering that we did not directly statistically compare males and females in any other analysis throughout the manuscript.

Figure 2. Total body weight loss (in % of baseline) compared between male and female animals expressing activating or inhibiting DREADD receptors in all MC3R cells. Body weight lost were plotted as percentage after 16h fasting in comparison to the body weight at injection. Two-way ANOVA were conducted.

5) I would make it abundantly clear that the DREADD experiments were performed in all MC3R-expressing cells, beyond just those in the PVT. Although they don't claim they are just manipulating PVT MC3R neurons, it can be interpreted that way based on the flow of the manuscript.

We have added text to make this clearer. It was never the intention to claim the MC3R mediated DREADD expression was PVT specific, so we have added textual cues to reduce this notion. We have also added a statement in the discussion addressing the limitation of this model.

Discussion lines 437-442:

'Indeed, one limitation of the model in our study is the expression of the DREADD receptors in all MC3R-positive cells throughout the body as driven by the MC3R-Cre model. However, as no difference in food intake itself was uncovered, it is likely that the MC3R effects are occurring outside the ARC. Subsequent studies will need to be performed to determine the possible contribution of peripheral MC3R activation as well as the specificity of PVT-MC3R cells in mediating this behavior.'

6) Related, a major question remains what would be the effect on feeding/body weight if manipulations were specific to MC3R expressing PVT neurons.

We thank the reviewer for the comment. We are also interested in investigating the PVT specific effects, and we propose performing this experiment in a PVT specific DREADD model.. However, due to the ethical restrictions to animal experimentation in Germany, we were unable to perform the proposed experiment within the current study. We intend to pursue this further if we are able to successfully acquire grant funding to support such experiments.

Interestingly, evidence from the literature supports the hypothesis that the PVT is involved in approach-avoidance behavior, especially in the context of stressors and food seeking (Engelke et al Nat Communications 2021 <https://pubmed.ncbi.nlm.nih.gov/33947849/>). Further, work by Betley et al. shows that specific manipulations of AgRP projections to the PVT are sufficient to drive feeding. Given our result showing the high expression of MC3R in this region, and previous literature showing no expression of MC4R here, it is likely that this change in feeding behavior is linked to alterations in MC3R activation in the PVT. However, these questions remain open to be explored and we look forward to continuing work in this direction.

7) It would be good to show some demonstration that CNO activates (hm3dq) or inhibits (hm4di) MC3R cells either via Fos staining or electrophysiology. Images of the expression of these DREADDs should also be shown.

We have added images of the DREADD expression in the PVT as a supplemental figure 8 to the manuscript.

Figure 3. (Supplemental Figure 8) Representative image of MC3R; hM3DGq(eGFP) mouse brain in the region of PVT. GFP was labeled in green and DAPI in blue. Scale bar equals 100 µm for the left figure, and 20 µm for the right figure. Red arrows indicate representative GFP-labeled DREADD expressing neurons.

Below we have included, for the reviewer, images and analysis showing cFos activation in the PVT upon CNO injection. However, a small caveat here, is that the CNO injections for these images were performed 2 weeks after the fasting/refeeding paradigm, therefore we do not have a direct readout of the c-Fos activation state during fasting or in the refeeding phase.

Additionally, in the hM3D experiments, perfusion was 20 minutes after CNO or Saline injection. In subsequent analyses we observed an unexpected high level of c-Fos expression in males given saline (Figure A). In the subsequent hM4Di experiments (and after consultation with an expert in stress biology), we prolonged the time between injection and perfusion to 1 hour. This resulted in a much lower c-Fos signal in the Saline injected animals (Figure B and D). We ultimately did not observe any significant differences between the Saline and CNO groups.

Another caveat here, however, is that all activated cells expressing c-Fos are detected, and not just the MC3R-DREADD cells. Thus, if there is a slight difference in the level of cFOS activation in MC3R-DREADD this may not be detected as other c-Fos positive cells are also detected. Due to the differing injection times before perfusion, we have not included any of the hM3D or hM4D c-Fos data in the manuscript.

[Figure removed by editorial staff per authors' request]

8) Do the authors have a hypothesis on why acute modulation of MC3R neurons changes the amount of time interacting with the food hopper? Are they not eating more during Phase 1? Is this related to food anticipatory behavior or general arousal? Can these be tested to infer what these results may mean?

Thank you to the reviewer for raising this question. If it was general arousal, we would predict that the animals in general would show more movement in the entire cage, but their time and movement is concentrated to the feeding area and they show no differences in total distance traveled (Supplementary Figure 13). Given the previously published work about the role of the MC3R in food anticipatory behavior (for example Begriche et al 10.1111/j.1601-183X.2012.00766.x , Sutton et al 10.1523/JNEUROSCI.3615-08.2008 amongst others), we think that it likely that there is a strong link between activation of these neuronal circuits during fasting and subsequent hyperfixation on food in the refeeding phase, when no CNO activation is present.

Our food intake data was collected at 1, 4, 12 and 24 hours post refeeding, and thus we do not have the food intake in the first 15 minutes (Figure 5D and Supplementary Figure 9C). However, given that there is no difference in food intake at any of the measured time points, we do not think the animals are eating more food. Instead we think it specifically has to do with food attentive behavior, and we are currently planning future studies to address this hypothesis.

We discussed this point in the text in lines 428-433

“This increased interaction with the food hopper is interesting given the documented role of MC3R in food anticipatory activity (FAA). In normal animals, an increase of locomotor activity is noted before the presentation of food, however, MC3R deficient animals show a significant defect in FAA (75, 76). Our data would suggest that hyperactivation of MC3R in the fasting period results in potential increases in food anticipatory behavior, food arousal or food attention. However, the exact behavioral changes and their underlying networks need further investigation”

9) I'd recommend shorting the Introduction.

We have taken the reviewers comment into consideration and have reduced the introduction to prioritize the focus on fasting, the PVT and the MC3R.

Reviewer #2 (Comments to the Authors (Required)):

10) This study examined the impact on MC3R expressing neuron activity by fasting during day or night phased and then the impact of increase or reduction in MC3R neurons on behaviors including feeding or drinking. Although analyses and individual experimental designs are sound, this manuscript is difficult to follow without a core hypothesis or a key insight revealed.

Thank you to the reviewer for the critical comments. With the input from both reviewers we have adjusted the manuscript text to aid in the flow and understanding of the manuscript. We hope the reviewer will agree that it is now easier to follow.

11) This study suffers greatly from lacking a key question to be addressed. This reviewer is confused why the study was conducted. The goal is to study MC3R or MC3R neuron function? or to study circadian regulation of MC3Rs? Many components seem to be involved but there appears to be no coherency here.

In order to improve coherency, we have both streamlined the introduction to prioritize the focus on fasting, the PVT and the MC3R. We have also significantly edited the final introductory paragraph to emphasize our primary hypothesis and questions to be addressed in the study. The text is now as follows (Lines 89-97 of revised manuscript):

“To identify PVT neurons mediating aspects of energy balance, we systematically characterized the neuroanatomical distribution of the MC3R within the thalamus and show that it is highly expressed throughout the PVT. We also analyzed how this nucleus responds to two fasting conditions; overnight and daytime, by quantifying IEG protein expression and phosphorylation of a known signaling mediator, which we hypothesized would be reduced within MC3R positive neurons. Furthermore, we questioned whether perturbation of MC3R activity during fasting would result in changes to refeeding behaviors in a mouse model with DREADD mediated activation or inhibition. This work shows the effects of time of day and energy state on PVT-MC3R neurons and highlights behaviors related to feeding which are mediated by MC3R activity in both male and female animals.”

12) There seems to be a big gap between anatomical studies (c-Fos and pCREB) and DREADD studies. The former is on MC3R neurons in PVT; however, the latter is all MC3R neurons/cells in the whole body. Therefore it is difficult to integrate the results from these studies.

We thank the reviewer for raising this point. We have reformatted the layout of the introduction to focus more primarily on fasting itself, as this is the linking behavior between the two main sections of the manuscript. Thus, with an intention to understand how fasting results in changes in brain activity, we used two approaches to assess the effects of fasting and the potential integration of responses through the MC3R. This was performed using a neuroanatomical approach to assess changes in the PVT and a physiological and behavioral approach using DREADD modulation of MC3R cells. The anatomical analysis was chosen given recent evidence of the role of the PVT in feeding and integrating stress and arousal signals as well as our data here depicting the robust expression of the MC3R in this region. The behavioral analysis was chosen given the prior literature linking MC3R and fasting responses.

Thus, with an intention to understand how fasting results in changes in brain activity, we used two approaches to assess the effects of fasting and the potential integration of responses through the

MC3R. This was performed using a neuroanatomical approach to assess changes in the PVT and a physiological and behavioral approach using DREADD modulation of MC3R-expressing cells.

July 23, 2024

RE: Life Science Alliance Manuscript #LSA-2024-02754-TR

Dr. Rachel Nicole Lippert
German Institute of Human Nutrition
Junior Research Group Neurocircuit Development and Function
Arthur-Scheunert-Allee 114-116
Nuthetal 14558
Germany

Dear Dr. Lippert,

Thank you for submitting your revised manuscript entitled "Fasting-induced activity changes in MC3R neurons of the paraventricular nucleus of the thalamus". We would be happy to publish your paper in Life Science Alliance pending final revisions necessary to meet our formatting guidelines.

- please be sure that the authorship listing and order is correct
- titles in the system and on the manuscript file must match
- please add your main, supplementary figure, and table legends to the main manuscript text after the references section
- please make sure LSA's formatting guidelines align the manuscript sections: please separate the Figure legends and Supplemental Figure legends into separate sections
- please add an Author Contributions section to your main manuscript text
- please add a Conflict of Interest statement to your main manuscript text
- there are call-outs for Figure 1C-P, and this figure doesn't have these panels, and they are also not mentioned in the legend...please correct

LSA now encourages authors to provide a 30-60 second video where the study is briefly explained. We will use these videos on social media to promote the published paper and the presenting author (for examples, see <https://docs.google.com/document/d/1-UWCfbE4pGcDdcgzcmiuJI2XMBJnxKYeqRvLLrLS08s/edit?usp=sharing>). Corresponding or first-authors are welcome to submit the video. Please submit only one video per manuscript. The video can be emailed to contact@life-science-alliance.org

A. FINAL FILES:

B. MANUSCRIPT ORGANIZATION AND FORMATTING:

Sincerely,

July 29, 2024

RE: Life Science Alliance Manuscript #LSA-2024-02754-TRR

Dr. Rachel Nicole Lippert
German Institute of Human Nutrition
Junior Research Group Neurocircuit Development and Function
Arthur-Scheunert-Allee 114-116
Nuthetal 14558
Germany

Dear Dr. Lippert,

Thank you for submitting your Research Article entitled "Fasting-induced activity changes in MC3R neurons of the paraventricular nucleus of the thalamus". It is a pleasure to let you know that your manuscript is now accepted for publication in Life Science Alliance. Congratulations on this interesting work.

DISTRIBUTION OF MATERIALS:

Again, congratulations on a very nice paper. I hope you found the review process to be constructive and are pleased with how the manuscript was handled editorially. We look forward to future exciting submissions from your lab.

Sincerely,
